# Fecal Microbiota Transplantation May Represent a Good Approach for Patients with Focal Segmental Glomerulosclerosis: A Brief Report

**DOI:** 10.3390/jcm11226700

**Published:** 2022-11-12

**Authors:** Wenqiang Zhi, Xiaoli Yuan, Wenzhu Song, Guorong Jin, Yafeng Li

**Affiliations:** 1Department of Nephrology, Shanxi Provincial People’s Hospital (Fifth Hospital) of Shanxi Medical University, Taiyuan 030012, China; 2Core Laboratory, Shanxi Provincial People’s Hospital (Fifth Hospital) of Shanxi Medical University, Taiyuan 030012, China; 3School of Public Health, Shanxi Medical University, No.56 Xinjian South Road, Taiyuan 030001, China; 4Shanxi Provincial Key Laboratory of Kidney Disease, Taiyuan 030012, China; 5Academy of Microbial Ecology, Shanxi Medical University, Taiyuan 030001, China

**Keywords:** chronic kidney disease, focal segmental glomerulosclerosis, fecal microbiota transplantation, treatment, brief report

## Abstract

This is the first report of fecal microbiota transplantation (FMT) in patients with chronic kidney disease. The patient was subjected to focal segmental glomerulosclerosis (FSGS), with onset in April 2021. The main manifestation featured abnormal renal function and no proteinuria at the level of nephrotic syndrome. In May 2021, she showed biopsy-proven FSGS and was treated with glucocorticoid. However, after glucocorticoid reduction, the patient’s serum creatinine increased again, so she adjusted the dosage and continued use until now. In April 2022, the patient was prescribed the FMT capsules. After FMT, the renal function remained stable, urinary protein decreased, reaching the clinical standard of complete remission, and there was no recurrence after glucocorticoid reduction. Furthermore, the patient showed significantly decreased hyperlipidemia, triglyceride (TG) and cholesterol (CHO) after FMT. During FMT, the level of cytokines fluctuated slightly, but returned to the pre-transplantation level after three months. From this, we conclude that FMT is a potential adjuvant therapy for FSGS, and patients can benefit from improving renal function and dyslipidemia.

## 1. Introduction

Focal segmental glomerulosclerosis (FSGS) represents one of the common pathological types of chronic kidney disease (CKD), and the incidence is increasing annually [1]. The prognosis of the disease is relatively poor, with about 40% of patients developing end-stage kidney disease within 5–10 years [2]. At present, glucocorticoid is recommended by KDIGO guidelines as the first-line treatment of FSGS [3]. The recommended time of glucocorticoid therapy is 6 months. Patients with glucocorticoid resistance are treated with immunosuppressants, but this is only limited to patients with nephrotic syndrome, and no effective treatment for patients with a non-nephrotic syndrome is available.

Previous studies have shown that intestinal microbiomes are associated with important organs and systems of the host, such as the brain, kidney, circulatory system, immune system, etc. [4,5,6,7]. Intestinal microecological disorders can affect the host’s multi-system balance and health (such as causing kidney inflammation) [8]. In CKD, the interaction between microbiota and host is bidirectional. On the one hand, a high toxin level in the circulatory system of patients with CKD affects the composition and metabolism of intestinal microflora. On the other hand, the imbalance of intestinal flora may be responsible for the destruction of the epithelial barrier, and may eventually lead to increased host exposure to endotoxin, resulting in an increase in the levels of toxic components of p-cresol sulfate, indoleol sulfate and trimethylamine-N-oxide (TMAO) in circulation, which may accelerate the progression of CKD [6]. As such, dietary therapy, and the supplementation of probiotics and probiotics to improve intestinal microecology, are considered promising adjuvant therapies for CKD [9,10,11]. Fecal microbiota transplantation (FMT) refers to the transplantation of functional flora from healthy human feces into the gastrointestinal tract of patients to reconstitute functional intestinal flora for the purpose of treating intestinal and extra-intestinal diseases (such as recurrent clostridium difficile infection, inflammatory bowel disease, systemic lupus erythematosus and IgA nephropathy) [12]. The technology has driven the treatment of gut microecology from the past elimination of single microorganisms and the increase in beneficial flora in the gut, to the reconstruction of the whole gut at the microecology level [13]. As a comprehensive approach to improving intestinal microecology, FMT may benefit patients with CKD [14].

## 2. Methods

The patient was a 65-year-old woman with CKD3 performing FMT. The outpatient clinic found that her serum creatinine was elevated. The serum creatinine was 277 umol/L, eGFR 15.83 mL/min/1.73 m^2^, CKD4. The patient subsequently underwent renal biopsy. Pathology suggested focal segmental glomerulosclerosis, and the types of renal lesions were characterized by glomerular segmental mesangial proliferative lesions, globular abandonment (1/8), segmental sclerosis (1/8), moderate chronic interstitial lesions (40%), and moderate acute lesions (30%), as well as Kappa (−), Lambda (−), C4d (−), PLA2R (−), IgG (−), IgA (−), HBcAg (−) and HBsAg (−), which were relieved after treatment with glucocorticoid 30 mg/d. However, when the glucocorticoid gradually decreased to 7.5 mg/d, the serum creatinine gradually increased again, so the drug dosage was adjusted to 10 mg/d again. Until now, there has been no significant remission in the amount of urinary protein in the patient, and the patient has long been distressed by the side effects of glucocorticoids (full moon face, hyperlipidemia, etc.). The patient had a history of HBV infection and did not receive regular treatment before April 2021. At that time, HBsAg (+), HBeAb (+) and HBcAb (+) were examined upon admission, and the quantity of HBV-DNA was lower than the limit of detection, which is also the reason why the initial use of glucocorticoids was small, because the use of glucocorticoids may cause hepatitis B virus replication, and the patient was treated with “entecavir 0.5 mg once every other day” antiviral therapy. For the same reason, we ultimately did not use immunosuppressive agents such as cyclosporine A.

This patient showed onset with renal insufficiency. The serum creatinine was 277 umol/L, urine protein was 2 g/24 h and serum albumin was 38.09 g/L, which are not up to the standard of nephrotic syndrome. No genetic testing was performed, but she was initially effective for glucocorticoid therapy; immunohistochemistry showed HBcAg (−) and HBsAg (−). She was pathologically diagnosed as FSGS; we think that the subtype of FSGS should be defined as FSGS—undertermined cause (FSGS-UC) [15,16,17]. The patient was initially treated with glucocorticoid, but due to the presence of HBV infection, considering the risk of HBV virus replication, sufficient glucocorticoid therapy was not used initially. The initial response to glucocorticoid was good, the renal function was improved and the amount of urinary protein decreased after treatment. However, when the glucocorticoid gradually decreased to 7.5 mg/d, the serum creatinine increased again, and we had no choice but to adjust back to 10 mg/d, which has bene maintained until now. The renal function of the patient was stable, but the quantity of urinary protein remained >500 mg/d, which indicates partial remission. KDIGO guidelines recommend glucocorticoid therapy for 6 months [3], but the patient has been maintained at a low dose for more than 12 months; renal function deteriorated after glucocorticoid reduction, and the status of HBV infection could not be ignored under the conditions of the long-term use of corticosteroids. 

In April 2022, we carried out FMT for this patient. Donor screening and the preparation of FMT capsules were carried out by Dongyuan Yikang Company (www.dongyuanyikang.com, 2 April 2022). First of all, the donors were evaluated by questionnaire, including medical history, behavioral risk and current health status. Fecal and serological tests are then conducted on potential candidates to rule out infectious diseases and diseases associated with potential intestinal microecological disorders. After screening, 2–3% of people became qualified donors. The fresh feces collected from the donors was finally diluted, filtered and centrifuged by professional and technical personnel in the Dongyuan Yikang clean laboratory to produce FMT capsules. FMT capsules were given on day 1, day 8, and day 15 for one course of treatment. The patient took 20 capsules each time, three times in total. During the FMT, the patient did not add or reduce any medication (compared with before enrollment). We collected blood, urine and fecal samples of the patient one day before FMT, the next day after FMT, one month after FMT and three months after FMT. Hematological parameters, liver enzymes, renal function, electrolytes, urine routine, and 24 h urine protein were measured. We also used flow cytometry to detect the levels of immune cells and cytokines during and after FMT. Metagenomic sequencing was used to detect the changes in intestinal flora and metabonomics during this period. 

## 3. Results

One month after the completion of FMT, the amount of urinary protein decreased from 420 mg/d to 290 mg/d, and the creatinine clearance rate increased from 44.66 to 49.73. Therefore, we adjusted the dosage of hormone from 10 mg/d to 10 mg/ every other day, and three months later, urinary protein decreased to 240 mg, creatinine clearance remained at 48.22, and serum creatinine remained stable. According to the definition of KDIGO guidelines, the standard of complete remission was reached. Meanwhile, the patient had previously had hyperlipidemia; after FMT, the triglycerides (TG) decreased from 1.65 mmol/L to 1.13 mmol/L, and cholesterol (CHO) reduced from 5.22 to 4.32 mmol/L. The patient had a history of HBV infection. We also found that the level of serum prealbumin (PA) increased from 189.1 mg/L to 276.5 mg/L one month after FMT, and to 262.6 mg/L three months later. The relevant results are listed in Table 1.

During this period, the immune functions of the patient remained stable and there was no significant change. The immune cells of the patient were at the normal level before and after FMT, and remained stable. The levels of some cytokines changed slightly during microbiota transplantation (IL-5, IL-8, IFN-γ, TNF-α). However, three months after FMT, these cytokines returned to levels similar to those before FMT. The relevant results are listed in Table 2 and Table 3.

We also used metagenomic sequencing to detect the changes in intestinal flora before and after FMT. The results show that among the ten bacteria with the highest abundance, the abundance of Prevotella copri and Bacteroides uniformis increased significantly, while the abundance of Bacteroides eggerthii and Roseburia intestinalis decreased. Although a clear relationship between changes in these bacteria and improvements in the clinical condition of this patient has not been fully demonstrated, it is important to note that some of these bacteria are associated with lipid metabolism and the progression of CKD in previous studies. This correlation may be discussed further in our future work. The result is depicted in Figure 1.

## 4. Discussion

As far as we know, this is the first report of FMT in patients with CKD and pathological FSGS. This patient had a good response to glucocorticoid therapy; when treatment was initiated, the patient had an insidious onset, without massive albuminuria, and the pathology was mostly mild and segmental. Immunohistochemistry showed HBcAg (−) and HBsAg (−); genetic testing was not performed on this patient. Based on the clinical presentation, we considered this patient as having FSGS—undertermined cause (FSGS-UC). The patient did not show significant hypertension and had a high serum creatinine level, and we decided not to use ACEI/ARB drugs after comprehensive consideration, and instead to change to low-dose glucocorticoids. 

The direct benefits of the clinical indicators of the patient were a decrease in urinary protein, the stability of renal function, a decrease in hormone use, and an improvement in blood lipid and liver function. One month after the completion of FMT, the amount of urinary protein decreased from 420 mg/d to 290 mg/d. After reducing the dosage of the glucocorticoids, the amount of urinary protein did not increase, and remained below 0.3 g/d. We believe that this is a significant change. 

In terms of immune function, the immune cells of the patient did not change significantly during the period of FMT and the follow-up period. Some proinflammatory cytokines increased slightly during the period of FMT, but returned to the pre-FMT level during the follow-up period. We think that this may be a mild inflammatory reaction caused by FMT [18], and the corresponding results also show that there was no serious immune response in the host after FMT. It is also suggested that FMT through capsules is a relatively safe strategy; of course, more cases need to be included for further demonstration in the future.

Although the patient did not show massive proteinuria (>3.5 g/d) before FMT, the patient has been dependent on glucocorticoid use for a long time, and each glucocorticoid reduction will lead to an increase in urinary protein and serum creatinine. In addition to CKD, the patient had long-term hyperlipidemia; their blood lipids could only be controlled after drug treatment, and could not return to normal levels. However, after FMT, patients’ triglycerides and cholesterol were successfully returned to normal levels, which may be related to the increase in *Prevotella copri* and *Bacteroides uniformis*. In previous studies, it was found that individuals with a high proportion of *Prevotella* had lower cholesterol than other groups. Similar findings were found in *Bacteroides uniformis* [19], and it was shown in mouse experiments that it could reduce the concentrations of plasma cholesterol, triglyceride and blood glucose in mice with a high-fat diet [20]. However, the patient stopped taking lipid-lowering drugs one month after FMT, and the concentration of triglycerides and cholesterol increased again in the second month of follow-up after FMT. We believe that the explanation may be twofold: one is that the patient stopped using lipid-lowering drugs on their own, and the role of FMT is to enhance the effect of lipid-lowering drugs, which may be used as an adjuvant treatment for hyperlipidemia in the future. The second is that the effect of flora colonization after FMT is not stable, and FMT may be a treatment to reduce blood lipids, but the specific way, dose and treatment plan of FMT still need to be further discussed. In patients with previous hepatitis B virus infection and long-term use of antiviral drugs, after FMT, the level of serum prealbumin is higher than that before transplantation, suggesting that the liver function has improved, but the specific mechanism of this improvement has not been determined. 

In terms of microflora, the abundance of *Prevotella copri* and *Bacteroides uniformis* increased significantly after bacterial transplantation. Studies have shown that the abundance of *Prevotella* decreased in CKD patients, and the decrease in *Prevotella* abundance was related to urea excretion in urine, which once again suggests the damaging effect of intestinal microecological imbalance on the kidney [21]. Metabolically, for people who eat a fiber-rich diet, having Prevotella-rich intestinal microbiota contributes to intestinal health, weight loss and cholesterol levels, and *Prevotella copri* has potential benefits in glucose homeostasis and host metabolism [22,23]. As mentioned earlier, *Bacteroides uniformis* has been shown to reduce cholesterol and triglycerides in human dietary intervention studies and animal trials. Because this paper discusses a single case study, it is not clear whether the decrease in blood lipids in this patient is related to the changes in microflora after FMT, but it is observed that there is a certain parallel relationship between the changes in microflora and the changes in blood lipids. This suggests the potential application value of FMT in hyperlipidemia in the future. 

*Bacteroides eggerthii* has been proven to be a potential biomarker of CKD in bioinformatics studies, but the significance of its decrease or increase has not been fully clarified [24]. Some studies have pointed out that its abundance is higher in patients with atherosclerosis, and animal experiments have shown that it can reduce the inflammatory response to colitis mice [25,26]. *Roseburia intestinalis* has been shown to improve inflammatory bowel disease, and some studies have also pointed out that it is abundant in patients with colorectal cancer, while its role in CKD remains to be further explored [27]. 

There are previous reports on the use of FMT in IgA nephropathy [28,29], one of which used the same FMT capsules as this patient. Further exploration of the relationship between renal disease and intestinal microecology, and an exploration of the best indication for FMT in renal disease, is our next step.

## 5. Conclusions

In short, this is the first report on using FMT to treat FSGS. After using FMT capsules, patients receive many benefits, and the results suggest that FMT may be used as adjuvant therapy for patients with CKD and pathological FSGS. During FMT and follow-up, no significant immune response was observed, suggesting that it was well tolerated by the host. Of course, more patients still need to be included to further support these results. Whether FMT has a therapeutic effect on some other subtypes of FSGS patients also needs to be further verified, while the mode, course of treatment, follow-up time and long-term effects on the immune function of FMT still need to be further explored.

## Figures and Tables

**Figure 1 jcm-11-06700-f001:**
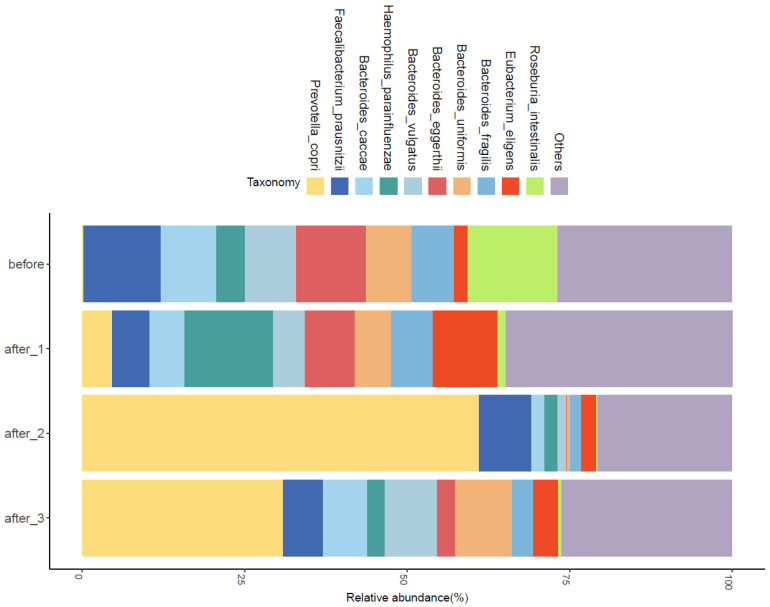
Changes in the abundance of the Top10 species-level bacteria before and after FMT. Before refers to before FMT, after_1 refers to after taking the first enterobacterial capsule, after_2 refers to after FMT completion, after_3 refers to one month after FMT completion.

**Table 1 jcm-11-06700-t001:** Changes in clinical indicators before and after FMT.

Stage	Urine Protein (g/24 h)	Scr (umol/L)	Ccr (mL/min)	TG (mmol/L)	CHO (mmol/L)	PA (mg/L)
Before FMT	0.42	131.4	44.66	1.95	5.22	189.1
After FMT	0.29	122.5	47.9	1.51	4.98	305.8
One month after FMT	0.26	118	49.73	1.13	4.32	278.6
Two months after FMT	0.29	117.2	50.07	2.37	6.46	276.5
Three months after FMT	0.24	121.7	48.22	2.28	6.28	262.6

Before FMT refers to the day before first drug administration. After FMT refers to the second day after the completion of FMT.FMT: Fecal microbiota transplantation;Scr: Serum creatinine; Ccr: Creatinine clearance; TG: triglyceride; CHO: cholesterol; PA: prealbumin.

**Table 2 jcm-11-06700-t002:** Immune cell changes before and after FMT.

Stage	lym	CD3%	CD3	CD4%	CD4	CD8%	CD8	CD4/CD8	B%	B	NK%	NK
Before FMT	1984	69.73	1383.44	51.33	1018.39	14.63	290.26	3.51	16.74	332	14.22	282
After FMT	1827	69.23	1265.00	49.57	906.00	16.07	294.00	3.08	17.29	316	12.66	231
One month after FMT	1899	67.86	1289.00	49.49	940.00	14.46	275.00	3.42	17.69	321	11.76	213
Three months after FMT	2178	74.07	1614.00	54.73	1192.00	16.56	361.00	3.30	16.84	410	8.81	214

Lym: lymphocyte, B: B cells; NK: natural killer cells.

**Table 3 jcm-11-06700-t003:** Cytokine changes before and after FMT (pg/mL).

Stage	IL-5	IL-4	IL-2	IL-10	IFN-α	IL-1β	IL-12 P70	IL-8	IL-17	IL-6	IFN-γ	TNF-α
Before FMT	3.401	0.096	0.824	0.17	0.005	9.637	0.668	89.38	1.956	3.065	31.442	31.71
After FMT	1.835	0.026	0.052	0.85	0.677	1.397	0.179	104.75	0.841	4.852	5.837	23.742
One month after FMT	1.835	0.155	0.052	0.104	0.434	0.981	0.331	86.40	1.956	3.065	24.297	22.178
Three months after FMT	4.861	0.118	2.031	1.031	1.234	0.63	0.374	92.14	1.602	3.612	0	56.598

## Data Availability

Raw data are available from the authors.

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
