# Peer review of "Fecal Microbiota Transplantation May Represent a Good Approach for Patients with Focal Segmental Glomerulosclerosis: A Brief Report"

_jcm, 2022, doi:10.3390/jcm11226700_

Round 1
Reviewer 1 Report
This manuscript is quite interesting.
But please add some as I wrote in the manuscript.

Author Response
Dear Professor,
We’re indebted to your valuable advice and guidance on our research. We have carefully looked into your comments, and accordingly made some changes in the article. Now, we have resubmitted our revised draft for your further suggestion and comment.
Our notes on the changes we have made in response to your suggestions are as follows.
- Line 32: We revised the unregulated statement in the first sentence of the introduction section,and modified ”FSGS” to “Focal segmental glomerulosclerosis (FSGS)”. It was our oversight and thank you for your conscientious review.
- Line 96: We detail the methodology used for FMT capsules.
- Line 101:This is our mistake. We have completed the expression of this sentence. Thank you for your patience.
- Questions about whether to give the patient other treatments during FMT: within three weeks of FMT, we did not increase or reduce any drugs that she had used before. One month after the completion of FMT, we reduced the dosage of glucocorticoids because of the good effect, and found that after this reduction of glucocorticoids, the patient's condition did not worsen, urine protein continued to decline, and renal function remained stable, We have added relevant statements (line 98-99) in the method section according to your suggestions, and also made relevant analysis on combined drug use in the result section (line105-110) and discussion section (line152-154).
- About the Table: We added an explanation, “Before FMT” refers to the day before the first drug administration, and “After FMT” refers to the second day after the completion of FMT.
- About Table 2 and Table 3: In our research plan, we did not follow up with the patient in the second month. Our follow-up time is the second day after the completion of FMT, one month after the completion of FMT, and three months after the completion of FMT. In the second month after the completion of FMT, the patient took the initiative to provide us with clinical-related data. Considering that the patient's test and the tests conducted in our research were all done in the Department of laboratory medicine and clinical testing laboratory of Shanxi Provincial People's Hospital, we added the data of the second month after FMT in Table 1, Data are missing for this part of table2 and table3, because patients were not conditioned for immune cell and cytokine testing at that time.

Reviewer 2 Report
This is a very interesting paper that offers insight into the pathogenesis and offers potential new treatments for a tough clinical disease. However, I suggest some clarifications to the paper:
-Line 55 states the patient has CKD stage3 and the next line says CKD4
-Hepatitis B has been shown to cause FSGS. How did you rule out hepatitis B as a cause of FSGS? Was there staining on the biopsy?
-Please give follow-up hepatitis B serologies after the patient was treated with entecavir. Hepatitis B-induced FSGS has been shown to improve after antiviral treatment, which may be the cause of this patient's improvement.
-KDIGO guidelines for maladaptive FSGS recommend not giving steroids. Please explain why this patient was treated with steroids if they were considered to have maladaptive FSGS.
-The response to steroids argues against maladaptive FSGS and favors primary FSGS as a cause. Please address this in the discussion.
-Was the patient treated with a renin-angiotensin inhibitor such as an ACE or ARB? This is considered first-line treatment for maladaptive FSGS and can affect urine protein excretion.
-Fecal microbiota transplant has been shown to change renin, ACE, and angiotensin levels in mouse models. Were renin, angiotensin, or aldosterone measured in this patient before or after transplant. This would make a much stronger study.
-Please address how the changing glucocorticoid dosing could affect the inflammatory cytokines you are reporting. How much is attributable to FMT and how much to the steroids?
Author Response
Dear Professor,
We’re indebted to your valuable advice and guidance on our research. We have carefully looked into your comments, and accordingly made some changes in the article. Now, we have resubmitted our revised draft for your further suggestion and comment.
- Line 55: We modified the statement that may be misleading here. The patient was judged as CKD4 according to eGFR at the beginning of the disease. After glucocorticoid treatment, her eGFR was improved. During FMT, we evaluated that the patient was at CKD3.
- The patient underwent immunohistochemical staining, and the results showed Kappa (-), Lambda (-), C4d (-), PLA2R (-), IgG (-), IgA (-), HBcAg (-), HBsAg (-). We believe that secondary FSGS caused by hepatitis B virus could be excluded.
- We reviewed relevant published papers, but unfortunately, we didn't find the evidence that hepatitis B virus can cause FSGS. Based on the immunohistochemical findings, we believe that hepatitis B-induced FSGS can be excluded in this patient.
- Thank you very much for this concern. There is indeed some difficulty in distinguishing primary FSGS from secondary FSGS. The patient did have some remission after treatment with glucocorticoids, which seems to be more inclined to primary FSGS. However, the amount of urine protein in this patient did not reach the level of nephrotic syndrome, albumin was not significantly reduced. Edema is also not obvious, so we initially considered that the patient did not meet the primary FSGS, and the patient is not currently undergoing genetic testing, and it is impossible to determine whether it is hereditary FSGS, so we are more inclined to believe that the patient was subjected to "maladaptive FSGS".
- Regarding the issue of ACEI / ARB versus glucocorticoids, although the KDIGO guidelines recommend the use of ACEI / ARB for maladaptive FSGS, we did not use them in this patient at the outset, and we considered the following reasons: 1. This patient had no obvious hypertension, renal vascular ultrasound showed high resistance of both renal arteries, and SPECT examination showed that the perfusion of both renal arteries was approximately normal; 2. Blood creatinine 277umol / L at onset in this patient, we must consider ACEI / ARB may cause further deterioration of renal function. Therefore, we finally chose the strategy of glucocorticoid therapy.
- Thank you very much for your guidance and advice, however, we did not measure renin, angiotensin, or aldosterone before and after FMT, which we will add in our follow-up study.
- Regarding the issue of immune cells and cytokines, we adjusted the dose of glucocorticoids one month after FMT, so we believe that the results Before FMT, After FMT and One month after FMT in our study are more attributable to FMT. The data from three months after FMT, we believe to be more due to the adjustment of glucocorticoid dosage, but it cannot be ruled out that it is the result of the combination of glucocorticoid dosage and the long-term effects of FMT.

Round 2
Reviewer 2 Report
The changes made in this draft have significantly improved the paper.
I have two remaining suggestions:
I recommend the authors use the classification of FSGS-undertermined cause (FSGS-UC) for this patient based on the KDIGO guidelines. This recommendations is for 2 reasons - first, the authors give no conclusive evidence for nephron loss in this patient that would be the cause of maladaptive FSGS; second, the patient responded to glucocorticoids which is not expected with the pathology of maladaptive FSGS but possible with FSGS-UC.
Please include the negative hepatitis stains from the biopsy definitively showing hepatitis B is not present in the podocytes. Here are 2 references indicating hepatitis B can cause FSGS so I think the paper should make clear how hepatitis B was ruled out as a cause of FSGS. Refs: (1) J Nephrol. 2005 Jul-Aug; 18(4):433-5. (2) Nephrol Dial Transplant. 2011 Jan;26(1):371-3.
Author Response
Dear Professor,
We’re indebted to your constructive advice and guidance on our research. We have consulted relevant literature and discussed it, and revised it according to your opinions.
First, we reevaluated the clinical subtype of this patient;
Second, thank you very much for the references, we added the immunohistochemical findings and discussed how to exclude secondary FSGS.